# Head and Neck Paragangliomas—A Genetic Overview

**DOI:** 10.3390/ijms21207669

**Published:** 2020-10-16

**Authors:** Anna Majewska, Bartłomiej Budny, Katarzyna Ziemnicka, Marek Ruchała, Małgorzata Wierzbicka

**Affiliations:** 1Department of Otolaryngology, Head and Neck Surgery, Poznan University of Medical Sciences, 60-355 Poznań, Poland; otosk2@gmail.com; 2Department of Endocrinology, Metabolism and Internal Diseases, Poznan University of Medical Sciences, 60-355 Poznań, Poland; bbudny@ump.edu.pl (B.B.); kaziem@ump.edu.pl (K.Z.); mruchala@ump.edu.pl (M.R.)

**Keywords:** pheochromocytoma, paraganglioma, head and neck neoplasms, head and neck tumors, genetic syndromes, mutations

## Abstract

Pheochromocytomas (PCC) and paragangliomas (PGL) are rare neuroendocrine tumors. Head and neck paragangliomas (HNPGL) can be categorized into carotid body tumors, which are the most common, as well as jugular, tympanic, and vagal paraganglioma. A review of the current literature was conducted to consolidate knowledge concerning PGL mutations, familial occurrence, and the practical application of this information. Available scientific databases were searched using the keywords head and neck paraganglioma and genetics, and 274 articles in PubMed and 1183 in ScienceDirect were found. From these articles, those concerning genetic changes in HNPGLs were selected. The aim of this review is to describe the known genetic changes and their practical applications. We found that the etiology of the tumors in question is based on genetic changes in the form of either germinal or somatic mutations. 40% of PCC and PGL have a predisposing germline mutation (including *VHL, SDHB, SDHD, RET, NF1, THEM127, MAX, SDHC, SDHA, SDHAF2, HIF2A, HRAS, KIF1B, PHD2,* and *FH*). Approximately 25–30% of cases are due to somatic mutations, such as *RET, VHL, NF1, MAX*, and *HIF2A*. The tumors were divided into three main clusters by the Cancer Genome Atlas (TCGA); namely, the pseudohypoxia group, the Wnt signaling group, and the kinase signaling group. The review also discusses genetic syndromes, epigenetic changes, and new testing technologies such as next-generation sequencing (NGS).

## 1. Introduction

Pheochromocytomas (PCC) and paragangliomas (PGL) are rare neuroendocrine tumors originating from either adrenomedullary chromaffin cells (PCCs); sympathetic ganglia of the thorax (T-PGL); or abdominal (A-PGL), pelvic, or parasympathetic ganglia in the head and neck (HNPGL) [1,2]. They are referred to collectively as PPGL. PCCs typically secrete one or more than one catecholamine: epinephrine, norepinephrine, and dopamine [1], while PGLs in most cases are non-secretory [1,3,4,5]. PCC represent 80% to 85% of chromaffin-cell tumors, and PGL represent 15% to 20% [6]. These tumors are characteristically well-vascularized and typically benign; nonetheless, roughly 10–15% may metastasize to the lungs, bone, liver, and lymph nodes. They most frequently occur between the third and sixth decades of life and present more commonly in women [7]. HNPGL can be categorized into carotid body tumors, which are the most common, as well as jugular, tympanic, and vagal paraganglioma. Other rare locations include the larynx, thyroid gland, parathyroid gland, nose, paranasal sinuses, parotid gland, or orbit [8]. PGL have also been described in the urogenital system, in the spermatic cord in particular [9]. Clinical symptoms vary according to the location and size of the tumor. Carotid body tumors typically produce a painless, slow-growing neck mass [10,11] that may eventually cause dysphagia and cranial nerve disorders. In contrast, pulsatile tinnitus and conductive hearing loss are characteristic of tympanic paraganglioma [12].

Neuroendocrine tumors show the highest degree of heritability in all neoplasms (approximately 40–50%) [13,14,15,16,17]. The first reports of the familial occurrence of PGL date from 1933, when carotid paragangliomas were first described by Chase [18,19]. In recent years, it has been confirmed that more than one-third of these tumors are genetically determined [20]. Today, the planning of further treatment considers family history, the extent and location of the tumor, its genetic origin, and the molecular pathways involved, especially as genetic testing becomes increasingly available and consistently improves the efficacy of therapy [3]. When a mutation is detected in a susceptibility gene such as *VHL*, *SDH,* or the recently discovered *MDH2*, a search for common co-occurring tumors is indicated [20,21]. Mutation in the SDHB subunit is also associated with the risk for malignancy and worse prognosis [3,10,22,23]. In 50% of patients with metastatic disease, a mutation in the *SDHB* gene was found. In the remaining 50% of cases, the genetic factors of the malignancy are still unidentified [23]. With this knowledge, genetic testing of PGL and the testing of first-degree family members should be routinely implemented to diagnose low-grade tumors [24]. Therefore, we aim to comprehend and conclude the most recent knowledge surrounding mutations in PGL, family occurrence, and their practical application based on the current literature and the paradigm of diagnostics.

## 2. Results

The outcomes are presented in the form of a literature review, structured by thematic subsections concerning the classification of head and neck paragangliomas with regard to genetic and molecular changes (based on 21 papers), as well as elucidation of genetic syndromes (based on 19 publications). Moreover, the review presents new methods as they pertain to the investigation of these tumors, such as investigation of epigenetic patterns or the application of new advanced molecular tools like next-generation sequencing (NGS) (based on five publications).

The details concerning the content of the presented articles (materials, methods, and conclusions) are presented in Table 1.

### 2.1. Classification Based on the Genetic and Molecular Background

Germinal mutations occur in the germ line and are passed on to all cells of the developing body [34]. A germline predisposing mutation is found in approximately 40% of PCCs and PGLs in one of at least 12 genes (*VHL, SDHB, SDHD, RET, NF1, THEM127, MAX, SDHC, SDHA, SDHAF2, HIF2A, HRAS, KIF1B, PHD2, FH*). The second type of genetic alteration is classified as somatic. These occur later in life, affecting only a single cell of a particular tissue, and give rise to the development of a specific neoplasm. Somatic mutations of *RET, VHL, NF1, MAX*, and *HIF2A* account for 25–30% of these tumors [13,16,23,32,33,35,36].

PGLs are classified into three clusters by the Cancer Genome Atlas (TCGA) on the basis of molecular, cytogenetic abnormalities, and specific single-nucleotide causative mutations, which led to the development of PPGLs. Moreover, contributing genes are grouped according to their biological activity—namely, the pseudohypoxia group, the Wnt signaling group, and the kinase signaling group. This division into groups with different clinical, imaging, molecular, and biochemical features allows for the personalization of patient care as well as the development of new screening and treatment guidelines [14,35,37,38].

The pseudohypoxia group can be further divided into two subgroups. The first comprises tricarboxylic acid cycle (TCA)-related factors concerning 10–15% of PPGLs. This group includes germline mutations in succinate dehydrogenase subunits *SDHA*, *SDHB*, *SDHC*, *SDHD* or *SDHAF2* (SDHx)—succinate dehydrogenase complex assembly factor 2, and FH (a second enzyme in the TCA cycle). The second subgroup encompasses *VHL/EPAS1*-related genes and accounts for 15–20% of PPGLs [14,35,37,38,39,40].

Activation of hypoxia inducible factors (HIFs) is a mutual characteristic for this cluster. HIFs are released in physiological response to cellular hypoxia. A pseudo-hypoxic state is caused by the presence of abnormal, mutated *VHL, SDH, EGLN1*, and *HIF2A* genes. The effect of this is constant activation of HIF pathways in the cell despite normal oxygen levels. This condition causes epigenetic changes in HIF target genes, which affects many processes including proliferation, angiogenesis, migration, apoptosis, and invasion. These events may all contribute to PPGL formation [19,35,38,41,42,43,44].

The Wnt signaling cluster is another group that are, in particular, triggered by somatic mutations in the *CSDE1* gene or somatic gene fusions which affect the *MAML3* gene. This results in the activation of Wnt and Hedgehog signaling pathways. Patients with sporadic PPGLs (5–10% of all PPGLs) are grouped here. Many developmental processes such as proliferation, cell polarity, adhesion, or differentiation are regulated by the Wnt pathway. As a result, these tumors are considered more aggressive, recur significantly, and are often prone to metastases [14,31,37,38,39,45].

The kinase signaling cluster (50–60% of PPGLs) includes germline or somatic mutations in *RET, NF1*, *MAX, HRAS*, and *TMEM127* genes [14,37]. The RAS/MAPK and PI3/AKT signaling pathways are enabled due to *RET* proto-oncogene activation or *NF1* tumor suppressor inactivation, resulting in tumor formation. In contrast, *TMEM127* mutations trigger the mTOR pathways. Another mechanism includes deactivation of the *MAX* suppressor gene, causing an abnormally elevated expression of cofactor *MYC* (proto-oncogene), resulting in the formation of PPGLs [14,38,39,40,41,43,44].

Several genetic syndromes are associated with PPGL: Multiple endocrine neoplasia type 2 (MEN2), Neurofbromatosis type 1 (NF1), Von Hippel–Lindau (VHL) disease, and Hereditary paraganglioma syndrome (PGL 1, PGL2, PGL3 and PGL4) [46,47].

HNPGL are very rare in NF1, MEN 2, and VHL patients. Rather, they display a predisposition toward the development of PCCs.

### 2.2. Genetic Syndromes

HNPGL are a solid manifestation in hereditary paraganglioma syndromes. They are caused by mutations in the succinate dehydrogenase (SDH) complex, which is necessary for the mitochondrial electron transport chain and ATP generation. This compound is composed of four subunits (A-D) with SDHAF2 stabilizing the entire complex. Subunits B, C, and D are strongly correlated with PCCs and PGLs [8,12,14,26,35,42,46,47,48].

PGL1 syndrome is an autosomal dominant disease linked to HNPGLs. It is correlated with inactivating mutations of the *SDHD* gene localized on chromosome 11q23. PCCs and sympathetic PGLs occur in 40% of cases, and bilateral or multifocal tumors are present in approximately 74% of patients. Though these tumors are typically not malignant, they have a tendency toward recurrence [14]. *SDHD* mutations are also associated with maternal genomic imprinting. Tumors are more likely to develop in children if the father is affected or a mutation carrier himself. If the mutation is inherited from the mother, it is inactivated but still genetically transmitted [8,12,15,35,41,47,49].

PGL4 syndrome also arises from a mutation with an autosomal dominant mode of inheritance, is responsible for inactivating the *SDHB* gene located on 11p35. In this condition, the following symptoms are reported: sympathetic extra-adrenal PGLs, PCCs, and HNPGLs. In up to 70% of all cases of PGL4 syndrome, the tumors are malignant [13]. PGLs typically produce catecholamines such as dopamine and norepinephrine, and only 10% of *SDHB* mutated tumors are biochemically silent; however, the clinical consequences are generally the result of significant mass effect rather than catecholamine excess. Typical tumor localizations include the abdomen and the mediastinum. The *SDHB* gene mutation increases the risk of renal cell carcinoma, gastrointestinal stromal tumor (GIST), and breast and papillary thyroid carcinoma, and while patients with metastatic disease should be routinely tested for the presence of the predisposing *SDHB* mutation, there are no guidelines regarding the screening of asymptomatic *SDHx* gene mutation carriers. Experts do suggest annual biochemical screening for PCC/PGLs from between the ages of five and 10, as well as full-body MRI screening for all associated tumor types every 2–5 years [8,12,14,35,41,47,49].

PGL3 syndrome is caused by an *SDHC* gene mutation located on 1q21-q23 and is inherited in an autosomal dominant pattern. PGL3 is associated with the occurrence of benign HNPGL, sympathetic PGL, and PCC and is typically multifocal. Metastases of these tumors is exceedingly rare [8,25,42,47,49,50].

Mutations in the *SDHAF2* gene have also been recently reported. *SDHAF2* mutation results in a rare type of familial paraganglioma syndrome that leads to HNPGL, but only in the children of a father who is a carrier of the defective gene. This syndrome is transmitted in an autosomal dominant manner, and usually manifests in the third decade of life. Genetic screening of *SDHAF2* mutation is crucial in patients with HNPGL with suspicious family history, young age of onset, or multiple tumors and have already tested negative for *SDHB, SDHC,* and *SDHD* mutations [27,28,29,37,46,47].

### 2.3. Epigenetic Patterns in HNPGL

Epigenetic changes are gene modifications that do not change the DNA sequence but affect gene activity. Most often the changes include methylation—the addition of a methyl group to the DNA strand—which results in the switching off or silencing of the gene and subsequent altered protein production. Other types of epigenetic modification include acetylation, phosphorylation, ubiquitylation, and sumoylation. Some of these changes can be inherited [51]. However, the most frequent of all epigenetic markers in DNA is cytosine methylation. This change in the human genome is referred to as “CpG methylation” or “DNA methylation” [52]. Inactivation of tumor-suppressor genes (TSGs) is caused by overall DNA hypomethylation and hypermethylation of CpG islands located in the closest vicinity of the promoter. Tumorigenesis of HNPGL is not yet fully explained, and the search for new genetic as well as epigenetic changes is ongoing.

In a study by Chen et al. [22], the methylation status of a panel of TSGs (*p16, HIC1, DcR1, DcR2, DR4, DR5, CASP8, HSP47, MGMT*, and *RASSF1A*) has been determined and compared in HNPGLs with and without SDH mutations. A correlation between the methylation index (MI) and the presence of germline mutations was observed. Six out of 10 TSGs showed frequent methylation: HIC1 and those involved in the apoptosis pathway DcR1, DcR2, DR4, DR5, and CASPS8. More frequent methylation in SDH-related HNPGLs compared to non-mutated analogues was observed in four analyzed TSGs (CASPS8, HIC1, DcR1, and DcR2). 

### 2.4. Next-Generation Sequencing (NGS)

Most of the studies conducted as of today have utilized conventional Sanger sequencing. Next-generation sequencing (NGS), in contrast to Sanger sequencing, enables broader and more accurate sequencing, leading to the detection of mutations in multiple genes. This technology allows for sample multiplication and also increases capacity and effectiveness, as well as reducing costs. Therefore, the use of NGS could provide the opportunity to test all patients at risk, rather than just a few selected targets [36]. It may provide a better understanding of the crucial role of the mutations acquired on various level of disease development, as well as those underlying the carcinogenesis of HNPGLs [53]. Luchetti et al. [30] analyzed 50 “mutation hotspot” variants in PCC and PGL using NGS in 20 patients with HNPGL and 85 patients with PPGL. The authors identified mutations in *HRAS* (7.1%), and *BRAF* (1.2%) as well as for *TP53* in 2.35% of cases. In the group of PPGL tumors with identified hereditary mutations (21 cases), *HRAS, BRAF*, and *TP53* genes were not mutated. It was concluded that the occurrence of *HRAS/BRAF* mutations predominates in sporadic PPGL (8.9%) but is inconsequential for inherited PPGL.

## 3. Materials and Methods 

This study assumes a review of world scientific literature. An online search was conducted using the scientific databases PubMed and ScienceDirect applying the key words head and neck paraganglioma and genetics. The first resulting article in PubMed dated from 1981 and from 1996 in ScienceDirect. Over the last 10 years, the number of articles on the subject has doubled. While this review considers articles from the last 20 years, over 85% of them were published in the last 10 years. Detailed data concerning the number of articles in each year are presented in Figure 1.

In total, 274 articles containing the indicated keywords were found in PubMed and 1183 in ScienceDirect. Of these, only those from the last 20 years reporting genetic changes in head and neck paraganglioma were selected.

## 4. Conclusions

The conclusions of this review are based on the entire overview of the literature and may prove useful for the improvement of diagnostic and therapeutic schemes surrounding PCCs and PGLs. According to the article “Recommendations for Somatic and Germline Genetic Testing of Single Pheochromocytoma and Paraganglioma” [54], the study of germline DNA should be prioritized in head and neck paraganglioma and thorax paraganglioma. A strong recommendation for genetic testing—somatic as well as germline mutations, regardless of the age at diagnosis—is indicated. It is also strongly recommended even in patients with a negative family history, especially if the lesions occur at a young age and are multifocal [55]. Genetic testing is very effective for predicting the incidence of metastatic tumors. Numerous authors [3,56,57] have demonstrated the variability in the *SDHB* gene, which leads to metastatic disease in 40% or more of patients. An agreement in the literature on the selection of mutations in HNPGL has been drawn, and encompasses the following genes: *SDHA, SDHB, SDHD, SDHAF2, SDHC, SDHB, VHL, FH, RET*. These should be routinely determined in PGL patients. Different combinations of these genes should be tested depending on the availability of a tumor sample or the performance of SDHB-immunohistochemistry (SDHB-IHC).

To conclude, the diagnostic schedule in PGL should include the collection of clinical data including epidemiology, family history concerning neoplasms, the course of the disease (e.g., tumor growth rate), and/or its relapses. Radiological evaluation of the tumor consisting of imaging and angiography (assessment of tumor size, vascularization, localization, position relative to other structures, presence of metastases) should also be considered. Furthermore, in light of the expanding knowledge of the genetic basis of this disease, genetic testing concerning causative alterations has become increasingly important. A multidisciplinary team consisting of an ENT specialist, a radiologist, an endocrinologist, a nuclear medicine physician, and a geneticist can qualify the patient on the grounds of such information for further treatment and the management of follow-up.

## Figures and Tables

**Figure 1 ijms-21-07669-f001:**
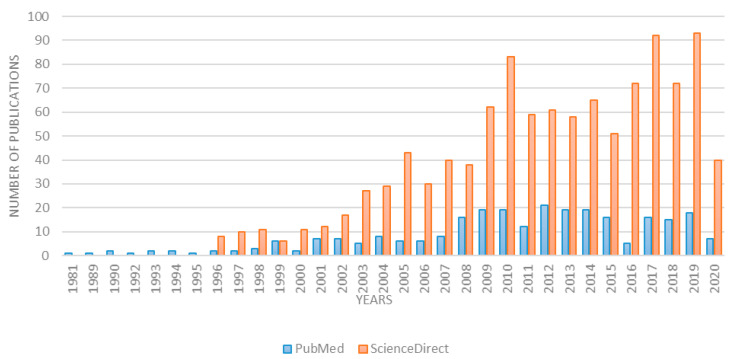
Number of publications according to key words head and neck paraganglioma and genetics.

**Table 1 ijms-21-07669-t001:** The table includes details concerning the content of the presented articles (authors, year of publication, number of patients in the study, reported genes, and most significant findings). Only data from original papers are included; no reviews are considered.

Author, Year	No. of Patients	Genes	Findings
Niemann et al. (2001) [25]	Five patients with histologically proven paraganglioma (single family members) and one patient (of this family) with imaging findings consistent with a PGL. 33 family members were clinically unaffected.	*SDHC* gene location	The disease locus in PGL3 was determined to be located at 1q21-q23.
Mannelli et al. (2009) [26]	501 patients with PCC and/or PGL 160 patients under 50 years of age whose DNA sequencing results revealed wild-type *RET*, *VHL, SDHB*, *SDHC*, and *SDHD* were subsequently analyzed for genomic rearrangements involving the *VHL* gene or one of the *SDH* genes.	*RET* *VHL* *SDHD* *SDHB* *SDHC* Genomic rearrangements (total deletion of the *SDHD* gene)	Detection of germinal mutations (such as *VHL, RET, NF1, SDHB, SDHC* and *SDHD*) in 32.1% of cases. From 100% in patients with associated lesions to 11.6% in patients with a single tumor. Genomic rearrangements were found in two of 160 patients (1.2%), both involving total deletion of the *SDHD* gene.
Bayley et al. (2010) [27]	443 patients with apparently sporadic PCC/PGL who did not have mutations in *SDHD*, *SDHC*, or *SDHB*. Examination of a Spanish family with HNPGL presenting with a young age of onset.	*SDHAF2*	No germinal (315 patients) or somatic (128 patients) mutations, and no germinal deletions of the *SDHAF2* gene were found. After pedigree analysis of a Spanish family with HNPGL a pathogenic mutation in *SDHAF2* was found that resulted in an amino acid substitution (p.Gly78Arg). The same mutation was noted previously in a Dutch kindred.
Kunst et al. (2011) [28]	57 family members.	*SDHAF2*	Establishing a correlation between HNPGL occurrence (based on phenotypic analysis) and *SDHAF2* mutation. The mutation carriers showed early onset of the disease and high levels of multifocality.
Casey et al. (2014) [29]	31 patients with confirmed PCC/PGL.	*TMEM127* *SDHAF2* *RET*	The occurrence of *TMEM127*, *SDHAF2* and *RET* mutations was found in patients without indications for genetic testing based on phenotypic evaluation.
Fishbein et al. (2015) [23]	Stage 1: whole exome sequencing on a discovery set of 21 patients with PCC/PGL. Stage 2: targeted sequencing of a separate validation set of 103 patients withPCC/PGL.	*NF1* *ATRX*	Mutations in *NF1* were detected in 42% of tumors. In 28% of *SDHB*-related tumors, deleterious variants of *ATRX* were found (PP119F1 p.W2275* and PP098F2 p.R2197H). ATRX protein was not detected in tumor cells by immunohistochemistry. The study found somatic mutation of *ATRX* in 12.6% of cases; 30% of them had truncating mutations and 69% missense mutations, classified as deleterious.
Luchetti et al. (2015) [30]	85 patients: PCC 60, PGL 5, HNPGL 20.	*HRAS* *BRAF*	Missense mutation was found in six cases (PCC = 6/60, PGL = 0/5, and HNPGL = 0/20) in *HRAS* in the hotspot region of codon 13 and 61. In one case of PCC, an activating *BRAF* mutation was found. In two patients a missense mutation was identified in the tetramerization domain of TP53 protein.
Fishbein et al. (2017) [31]	173 patients with PCCs/PGLs.	*SDHB*, *RET*, *WHL*, *NF1,* *SDHD*, *MAX* *EGLN1 (PHD2)*, *TMEM127*, *CSDE1*, *HRAS*, *EPAS1*, *MAML3,* *BRAF*, *NGFR*	27% of patients had germinal mutations (including *SDHB* 9%, *RET* 6%, *VHL* 4%, and *NF1* 3%). *SDHD*, *MAX*, *EGLN1* (*PHD2*), and *TMEM127* mutations were found in less than 2% each. *CSDE1* was identified as a somatically mutated driver gene complementary to the other four known drivers (*HRAS, RET, EPAS1*, and *NF1*). *MAML3*, *BRAF*, *NGFR*, and *NF1* fusion genes were discovered.
Bausch et al. (2017) [32]	972 unrelated patients without mutations in the classic PCC/PGL associated genes.	*SDHA*, *TMEM127*, *MAX*, *SDHAF2*	Six percent of patients were mutation carriers (including *SDHA*, *TMEM127*, *MAX*, and *SDHAF2*). 91% of patients had familial, multiple, extra-adrenal, and/or malignant tumors and/or had younger age of onset. Extra-adrenal tumors occurred in 48% of mutation carriers and in 79% of carriers with HNPGL.
Chen et al. (2017) [22]	37 patients with HNPGLs.	*SDHD* *SDHB* *SDHAF2*	*SDHD* gene mutations were found in: the Chinese founder mutation (c.3G>C, p.Met1Ile) in six cases, a missense mutation (c.284T>C, p.L95P) in one case, an in-frame deletion (c.278–280delATT, p.Y93S) in one case. A missense *SDHB* mutation (c.647A>G) and a nonsense *SDHAF2* mutation (c.130C>T, p.Gln44Ter) were found in two cases. Frequent methylation was observed in six of the TSGs tested (*HIC1, DcR1*, *DcR2, DR4, DR5*, and *CASPS8*). Four of them (*HIC1*, *DcR1*, *DcR2* and *CASPS8*) showed more frequent mutations in *SDH*-associated HNPGL than in non-mutated ones.
Calsina et al. (2018) [21]	830 patients with PPGLs, negative for the main PPGL driver genes.	*MDH2*	Twelve heterozygous variants of *MDH2* were found (five of the 12 were missense (41.7%), one synonymous (8.3%), four were located in the intronic region (33.3%), one was an in-frame deletion (8.3%), and one affected a donor splice-site (8.3%). Five of these were unreported variants. The study showed the functional impact of two variants (p.Arg104Gly and p.Lys314del) and suggests altered molecular function of two variants (p.Val160Met and p.Ala256Thr).
Ding et al. (2019) [33]	23 cases of multiple HNPGL.	*SDHD*, *SDHB,* *SDHC*, *SDHAF2*, *VHL*, *RET*	Family 1: 12 *SDHD* mutations (8 bilateral carotid body tumor (CBT) with 1 bilateral malignant CBT) Family 2: 3 *SDHD* mutations (1 bilateral CBT, 2 unilateral CBT) Family 3: 2 cases of *SDHD* mutations (vagus PGL and pheochromocytoma) Other patients: sporadic manifestations (5 cases *SDHD* gene mutation, 1 case *RET* gene mutation). Two novel mutations were found: c.387–393del7 mutation of *SDHD* gene and c.3247A>G mutation of *RET* gene. More frequent occurrence of *SDHD* mutations was found in patients and family members with multiple HNPGL.

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
