# Peer review of "Head and Neck Paragangliomas—A Genetic Overview"

_ijms, 2020, doi:10.3390/ijms21207669_

Round 1
Reviewer 1 Report
These authors have reviewed the data on genetic influences on paragangliomas, and summarise the recent literature and recommendations. It is generally well-written and succinct, but the English does require modification by a native English speaker.
I am not clear as to the reasons for the choice of the tabulated references. There is a vast literature on this subject, with a huge number of case reports or small series, so it is difficult to see a rationale for the chosen cited papers. Can the authors explain this and how they reached their conclusions?
They could also usefully note MDH mutations.
Author Response
Dear Reviewer,
Thank you for your comments and efforts to improve this paper. The selected articles were focused particularly on solid genetic changes in the head and neck paragangliomas. Papers discussing genetic changes leading to associated pheochromocytoma were removed. The applied corrections also included MDH variants in the article (line 62, the entire 11th row in the table). Additionally, several interesting and recent case reports were included (line 84, citation no. 9; line 237-239, citation no. 55). We considered this set of papers as most clinically relevant for the field and our manuscript, and on this basis we draw our conclusions. The work was proof read and edited by a native speaker. We hope that applied changes met your expectations.
Corrections are highlighted in the amended version of the manuscript.
Yours sincerely,
Anna Majewska
On behalf of the authors
Reviewer 2 Report
Thanks for having allowed me the opportunity to review this paper. It is a welcome addition to current literature. I can offer the following comments, overall minor in nature:
- the abstract is poorly informative. I suggest a deep revision of this section, in order to better present the aims and scopes, and the main findings of this review
- Materials and methods: this is definitely not a meta-analysis, but rather a narrative review based on a literature research. the Authors should clarify this.
- Table 1 maybe simplified, e.g. citation of papers can be shortened (first author et al). Furthermore, the "Findings" column should not just cite verbatim the main finding of each paper, but rather present a critical rewording
- I suggest the Authors to revise the entire paper in terms of editing and language
Author Response
Dear Reviewer
Thank you for your comments and efforts to improve this paper. I have adjusted the article according to all suggestions. The abstract section has been thoroughly revised (line no. 8-26). The main objectives, findings and conclusions were now implemented in abstract, so we think it is now more informative and fulfil its function. Description of materials and methods was revised, and now it’s clear that this is a review and not a meta-analysis (line no. 72-73). The citations in the table were simplified and the column describing major findings has been rewritten to conclude crucial outcomes (table 1 – page 4-8). The work was proof read and edited by a native speaker. We hope that applied changes met your expectations.
Corrections are highlighted in the amended version of the manuscript.
Yours sincerely,
Anna Majewska
On behalf of the authors